# Systematic Compounding of Ceramic Pastes in Stereolithographic Additive Manufacturing

**DOI:** 10.3390/ma14227090

**Published:** 2021-11-22

**Authors:** Soshu Kirihara

**Affiliations:** Joining and Welding Research Institute, Osaka University, 11-1 Mihogaoka Ibaraki, Osaka 567-0047, Japan; kirihara@jwri.osaka-u.ac.jp

**Keywords:** additive manufacturing, stereolithography, ceramic component, nanoparticle paste, dental crown

## Abstract

In this paper, stereolithographic additive manufacturing of ceramic dental crowns is discussed and reviewed. The accuracy of parts in ceramic processing were optimized through smart computer-aided design, manufacturing, and evaluation. Then, viscous acrylic resin, including alumina particles, were successfully compounded. The closed packing of alumina particles in acrylic pastes was virtually simulated using the distinct element method. Multimodal distributions of particle diameters were systematically optimized at an 80% volume fraction, and an ultraviolet laser beam was scanned sterically. Fine spots were continuously joined by photochemical polymerization. The optical intensity distributions from focal spots were spatially simulated using the ray tracing method. Consequently, the lithographic conditions of the curing depths and dimensional tolerances were experimentally measured and effectively improved, where solid objects were freely processed by layer stacking and interlayer bonding. The composite precursors were dewaxed and sintered along effective heat treatment patterns. The results show that linear shrinkages were reduced as the particle volume fractions were increased. Anisotropic deformations in the horizontal and vertical directions were recursively resolved along numerical feedback for graphical design. Accordingly, dense microstructures without microcracks or pores were obtained. The mechanical properties were measured as practical levels for dental applications.

## 1. Introduction

Stereolithographic additive manufacturing (STL-AM) has been performed to create industrial components of alloys and compounds [1,2,3,4]. Nanoparticles with different mechanical properties, thermal conductivities, electromagnetic responses, and biological compatibilities can be dispersed into photosensitive resin pastes as printing inks. A solid point is photochemically polymerized at the laser spot of an ultraviolet (UV) layer. One-dimensional (1D) fine wires of 50–100 μm in diameter are adjacently created along the scanning lines. Two-dimensional (2D) cross sections are voluntarily printed by crossover drawing, whereas three-dimensional (3D) solid structures can be freely fabricated by layer lamination and interlayer bonding [5,6]. Composite precursors should be successfully converted into full metal and ceramic objects through optimized dewaxing and sintering. Accordingly, computer-aided design, manufacturing, and evaluation have been continuously advanced [7,8].

Functional particles should be adequately selected to embody practical structures. Geometric distributions of material properties can perform spatial modulations of energy and mass transfers. Therefore, periodically arranged patterns with dielectric constants for electromagnetic wave modulations, ordered porous electrodes of solid electrolytes for effective energy generation, graded lattice scaffold distributions of biological ceramics for artificial metabolism promotions, and diagonally connected supports in bulky alloys for component weight reductions were precisely fabricated [9]. Monosized particles were dispersed at a 50% volume fraction. Elaborate micropatterns were obtained using a precise heat treatment. Dimensional reductions were measured at 20% linear shrinkage. Accordingly, reproducible reductions could be precisely offset via a numerical feedback of the shrinkage ratios for redesigns [10].

Considering the rapid fabrication of biological implants, ceramic dental crowns can be successfully processed by using computer aided design and manufacturing (CAD/CAM) [11]. Calcinated zirconia bodies with disc shapes are automatically cut and shaped by top–down processing. Crown precursors can be sintered into fine ceramic components with mechanical properties that satisfy the practical strengths required by dental bridges. Recently, zirconia crowns and connected bridges are systematically fabricated by using STL-AM [12,13]. In comparison, our research group has attempted to process alumina dental crowns by STL-AM using viscous resin pastes with dense particle dispersions [14]. Functional features could be formed on inner crown surfaces via bottom-up processing to realize rigid fastening with artificial dental roots. The translucent bodies could exhibit mechanical properties sufficient for single crown use.

Paste materials should be systematically compounded to achieve a strict accuracy. In this review, ceramic dental crowns with designed features for intraorally compatible occlusions will be demonstrated. Multimodal powders were dispersed at an 80% volume fraction. Linear shrinkages were reduced within 7% in the horizontal and vertical directions. On dental treatments, machined teeth should be covered with prosthetic appliances. The gap spaces between natural and artificial were intentionally adjusted at an average interval of 50 μm in all directions. Dental cements were tucked to provide sufficient adhesive strength and durability. Consequently, comparative investigations between virtual simulations of particle dispersions and real compounding of paste materials were discussed and reviewed along with the smart processing of alumina dental crowns with fine ceramic microstructures.

## 2. Procedures

### 2.1. Dispersion Modeling and Numerical Simulation

The dispersion profiles of solid particles in liquid matrices were simulated using the distinct element method (DEM) (EDEM, Cybernet Systems Co., Ltd., Tokyo, Japan) [15]. The individual particles were postulated as rigid spheres. Steric overlaps were allowed to consider multibody contacts. The interparticle forces were decomposed into normal and tangential directions. Both motion components were virtually expressed as the resultant movements of the springs and dampers connected in parallel. The contact times and restitution coefficients of collision impacts were adjusted along the spring elasticities and damping viscosities. A virtual slider was connected in series to the tangential component. The pose angles were calculated using frictional coefficients. The kinetic parameters of the virtual devices were entered according to the mechanical and rheological properties of solid particles and liquid matrices. Spatial distributions of dispersed particles were equally partitioned by cubic boundaries. Consequently, the edge lengths were similar to the particle diameters. Contact determinations were partially executed in the center and adjacent cubic areas to save time. Particle spheres with diameter distributions were closely packed in the virtual space. The central coordinates of the individual spheres were obtained as numerical datasets.

Particle distributions in viscous fluids were automatically visualized using a computer graphics (CG) application (Fusion 360, Autodesk, Tokyo, Japan). Binary spheres with small and large diameters were randomly packed. Size distributions were determined stepwise according to real compounding ratios of the small and large particles. Volume fractions were calculated for individual dispersion profiles to optimize packing densities. Molecular film formations were assumed as surface buffer layers at particle contacts to estimate correction values. Laser beam expansions in particle dispersion models were simulated using a ray tracing (RT) application (LightTools, Cybernet Systems, Tokyo, Japan) [16]. Coherent light waves with modulated irradiation powers and times were introduced from a circular area, which was assumed to be a beam spot. Scattering and attenuation profiles were visualized according to refraction indices and absorption rates in composite media. The photo-reactable zones were numerically specified from the intensity distributions of electromagnetic wave energies. The irradiation power per unit area and time could be optimized for effective photopolymerization. Laser drawing parameters and layer lamination procedures were chosen as the experimental conditions in the STL-AM.

### 2.2. Slurry Preparation and Stereolithography Setting

Alumina particles with average diameters of 170 nm (Taimicron TM-DAR, Taimei Chemicals, Nagano, Japan) and 1.7 μm (AL-170, Showa Denko, Tokyo, Japan) were bimodally mixed with acrylic resin (KC1287, JSR, Tokyo, Japan) at volume percentage of 10% and 70%, respectively, using a mechanically degassing and dispersing machine (SK-350T, Shashin Kagaku, Kyoto, Japan). Functional groups as surfactant agents were chemically added to acrylic oligomers for factory production. Paste materials with a total volume of 500 mL were packed into cylindrical polyethylene vessels with an inner size of φ100 × 100 mm. Ten 5-mm alumina balls were simultaneously used for effective stirring. Air bubbles were centrifugally separated in revolving movements for active degassing. Particle coagulation was dynamically eliminated during rotating movements for homogeneous dispersion. The rotating and revolving frequencies were systematically adjusted to 700 and 300 rpm, respectively. The total times of the mixing procedures were stepwise extended for 300, 600, and 900 s. Ceramic particles revealed by ethanol blasting of the paste materials were observed by scanning electron microscopy (SEM) (JSM 6060, JEOL, Tokyo, Japan).

The rheological behaviors of the paste materials were characterized according to individual mixing times using a kinematic viscometer (KV) equipment (VT550, Thermo Fisher Scientific, Waltham, MA, USA). Mixed materials were interposed between the upper and lower metal disks. The viscous paste was inserted into an inverse conical gap with a slant angle of 1.0°. The minor excess portion was left protruded without removal. An upper conical disk with a diameter of 20 mm was mechanically rotated at 0.2 rad/s^2^ under constant acceleration and deceleration rates. Circular frequencies increased and decreased between the static and dynamic states from 0 to 10 rad/s. A lower flat disk with a diameter of 30 mm was connected to a torque meter for the dynamic measurements of resistance forces. All the procedures were automatically executed in air atmosphere at room temperature. The dynamic profiles of kinematic viscosities were described as hysteresis curves. The horizontal axis represents the shearing speed equal to the circular frequency of the upper disk, and the vertical axis represents the shearing stress calculated as the resistance force divided by the paste area on the lower disk. Sequential results were systematically obtained based on the paste mixing times.

### 2.3. Process Optimization and Component Evaluation

The manufacturing parameters were experimentally optimized by the systematic operations of an STL-AM, as shown in Figure 1. An aluminum alloy stage was descended for 1 mm from the initial level. A polyethylene syringe was linearly moved for 100 mm in the path length. A paste material with a volume of 10 mL was dispended from a nozzle with a hole diameter of 1 mm through air injections with pressure of 0.2 MPa. A Teflon-lined stainless-steel blade was mechanically drove at a speed of 50 mm/s to smoothly spread in a movement width of 100 mm. A UV laser beam with a wavelength of 355 nm was focused into 50 μm as the minimum spot size. Irradiation powers were systematically adjusted from 50 to 350 mW. A beam spot was scanned at a maximum speed of 3000 mm/s. Test patterns were drawn on the top surface, as shown in Figure 2. A round hole with an inner diameter of 500 μm was designed at the center of a square film with an edge length of 10 mm. The hole diameters and film thickness were measured to estimate dimensional tolerances and curing depths using a digital optical microscopy (DOM) equipment (VH-Z100, Keyence, Osaka, Japan).

Operating configurations in STL-AM were automatically generated using a CG application (Magics, Materialise, Yokohama, Japan). A dental crown model with polyhedrally approximated features was digitally sliced at optimized lamination pitches of layer thicknesses. The polygonal outlines of the cross sections were slightly shifted inside to offset dimensional tolerances. Raster patterns were intersected perpendicularly to fill the sections. All processes were sequentially executed using an operating application (Sezac, Shashin Kagaku, Kyoto, Japan). Then, the cross-sectional layers were solidified via laser drawing on the spread pastes. The parts’ dimensions were measured using a 3D scanner (SOL 3D, Scan Dimension, Alleroed, Denmark). The dimensional accuracy was evaluated through a comparison with the designed models. Obtained precursors were used for heat treatment in air atmosphere. The heating rate was controlled at 1.0 °C/min using thyristor inverters. Dewaxing temperatures and holding times were adjusted to 600 °C for 2 h. Sintering bodies were obtained at 1300 °C for 2 h. Total duration time of the heating, dewaxing, sintering, and cooling was 30 h. Afterward, the microstructures were observed using SEM. Plate specimens with dimensions of 20 mm × 5 mm × 1 mm were similarly fabricated. The mechanical strengths were measured by four-point bending tests for dental ceramic materials according to ISO6872, the micro-structural hardness was assessed by Vickers testing, and the relative densities were measured using the Archimedes method. The strength, hardness, and density were obtained from each specimen, which were arranged into groups of seven such that the highest and lowest measured values could be excluded and the average values calculated from the remaining five data.

## 3. Results and Discussion

Volume fractions for compounding ratios in a binary system of particle sizes were calculated stepwise through DEM simulations, as shown in Figure 3. The average sizes of the small and large particles were adjusted to 170 nm and 1.7 μm, respectively. Particle size variations of the selected alumina powders were entered as frequencies in the distribution specification tables. The closest packing of an 80% volume fraction was obtained at a compounding ratio of 0.875. The dispersion profiles were virtually plotted through a CG visualization, as shown in Figure 4. The particle features simulated spherical shapes according to the microscopic observations. Small particles were interstitially dispersed between large particles. The light propagations were systematically calculated through RT simulations, as shown in Figure 5. The laser beams were adjusted to a spot diameter of 50 μm, and intensity distributions were visualized on vertical cross sections. The photo-curable areas were estimated as light penetration limits. Furthermore, boundary surfaces were defined at contour lines of 0.1% for the transmission attenuation. A layer thickness of 100 μm in curing depth should be obtained by laser drawing at an irradiation power of 200 mW. The scanning speed was increased to 3000 mm/s as the maximum value. The alumina particles were observed via SEM, as shown in Figure 6a. Coagulated grains were interspersed through a short-time pasting under 300 s. Then, the impact frequencies between the ceramic balls and vessel walls were insufficient for fine pulverization. The particles were gradually dispersed according to the pasting times, as shown in Figure 6b. The coagulated particles are remained in the SEM view of Figure 6c. However, homogeneous distributions could be observed over a long period of time over 900 s. The rheological profiles were measured by a KV, as shown in Figure 7. Shearing stresses linearly increased and decreased for shearing speeds after a short pasting time of less than 300 s. The isolated resin from the coagulated particles behaved as a Newtonian fluid as shown in Figure 7a. In addition, hysteresis curves were steadily separated according to the pasting times, as shown in Figure 7b,c. The same profiles could be obtained through long periods of pasting over 900 s. Paste viscosities were dynamically modulated by the inertia friction between nearly stationary particles. Interactive dodging of actively dispersed particles could decrease the viscosity. The hysteresis property is called thixotropy. Accordingly, saturated rheological profiles should be similarly obtained for the same particle size dispersions through different mixing equipment and processing times.

In this study, the curing depth and dimensional tolerances were systematically measured using DOM, as shown in Figure 8. A beam diameter of 50 μm was used as the horizontal unit size for laser drawing, whereas a lamination pitch of 50 μm was used as the vertical unit size for layer stacking. The irradiation power was adjusted to 100 mW to realize a curing depth of 75 μm for interlamellar joining. A dimensional tolerance of 10 μm could be balanced with the offset value in the laser drawing. Lamination pitches and offset values were simultaneously optimized to establish processing times and part accuracies. A composite precursor of the dental crown was fabricated along the CG model using STL-AM, as shown in Figure 9a,b. Solid layers of 50 in total were continuously laminated for 40 min in the process time. The part accuracies were measured as ±5 μm in the horizontal and vertical dimensions, and the microstructures were observed using SEM. Alumina particles were showing homogeneous dispersions in the acryl matrix. Impressions of blowholes contaminations at paste spreading and delamination occurrences at layer bonding were not left in the microscopic field. The results show that the measured densities of 3.27 g/cm^3^ were in good agreement with the values calculated by the compound rule.

Furthermore, a ceramic crown was successfully fabricated through dewaxing and sintering, as shown in Figure 9c. Open cracks or delamination gaps were not observed on component surfaces. Dimensional images with internal and external features were detected using 3DS. Graphic models with designed and measured dimensions overlapped in the virtual space. A volume shrinkage of 20% was successfully obtained according to the volume fraction of ceramic particles in the composite precursor. Linear shrinkages for the horizontal and vertical directions were estimated to be 6.7% and 8.1%, respectively. Moreover, the cubic root of volume shrinkage intervened in the values of both linear shrinkages. Gravitational forces should compress treated objects during sequential dewaxing and sintering. The reproducibility of dimensional shrinkages was similarly verified on repetitive heat treatments for multiple samples. Graphical models were successively expanded for three dimensions to offset the volume and linear shrinkages. Consequently, the average part accuracy was improved to ±5 μm. Misfit fragments were detected in the designed and scanned crown models. The exclusive logical sum was automatically visualized using Boolean operations, and the inconsistent volumes were less than 1% compared to the whole model.

Microstructures were observed by SEM for the horizontal and vertical planes, as shown in Figure 10a,b, respectively. Equiaxed grains with an average diameter of 10 μm were homogeneously distributed, whereas micropores or cracks were not included in the shown images. The sound layer joining without delamination gaps was verified for the vertical stacking directions. Binary particles were effectively joined by surface diffusion. Grain growth could sufficiently expand over the layer boundaries. The crystal phase of α-alumina was detected as a peak pattern in the X-ray diffraction. However, residual carbon originating from acrylic resin was not observed as carbide impurities. Based on the Archimedes method, the relative density was estimated to be 99.8%. The average bending strength was 480 MPa for the plate specimens. The mechanical properties could exceed the practical criteria for a single ceramic crown at a compression strength of 160–240 MPa. The Vickers hardness of the crown surfaces was measured at 1600 HV. Dental glasses could be successfully coated on the crown surfaces, and the molten phase could penetrate into micro-steps, consequently forming layer laminations. Furthermore, color tone compensations for biomimetic textures can be demonstrated using pigment dispersion into glass layers.

## 4. Conclusions

In this study, ceramic dental crowns were successfully fabricated using STL-AM, nanoparticle dispersions in viscous pastes were graphically visualized using a DEM, and closed packing conditions were systematically optimized. In addition, intensity distributions from irradiation spots of UV laser beams were spatially simulated using the RT method, and the photochemically polymerized areas were compared numerically. Alumina particles with bimodal size distributions could be dispersed into acrylic resin at 80% of the total volume fraction. The curing depth and dimensional tolerances were precisely measured using DOM. The process conditions of layer drawing and lamination bonding were adjusted. The composite precursors were dewaxed and sintered at 600 °C and 1300 °C at heating temperatures at a holding time of 2 h. The results show that the relative density reached 99.8%, and the linear shrinkages were estimated to be 6.7% and 8.1% for the horizontal and vertical directions, respectively. Anisotropic deformation rates were recursively resolved using numerical feedback for graphical design. The mechanical properties were obtained at a practical level at an average bending strength of 480 MPa and Vickers hardness of 1600 HV. Consequently, a smooth surface with translucent textures could be realized. Ceramic pigments should be automatically distributed for biomimetic printing in color tones. In the future, dental implants of crowns and roots will be fabricated using other ceramic materials with high rigidity and biological compatibilities. Nonetheless, we believe that lithography paste compounding with close-packed filters is a key technology in this field.

## Figures and Tables

**Figure 1 materials-14-07090-f001:**
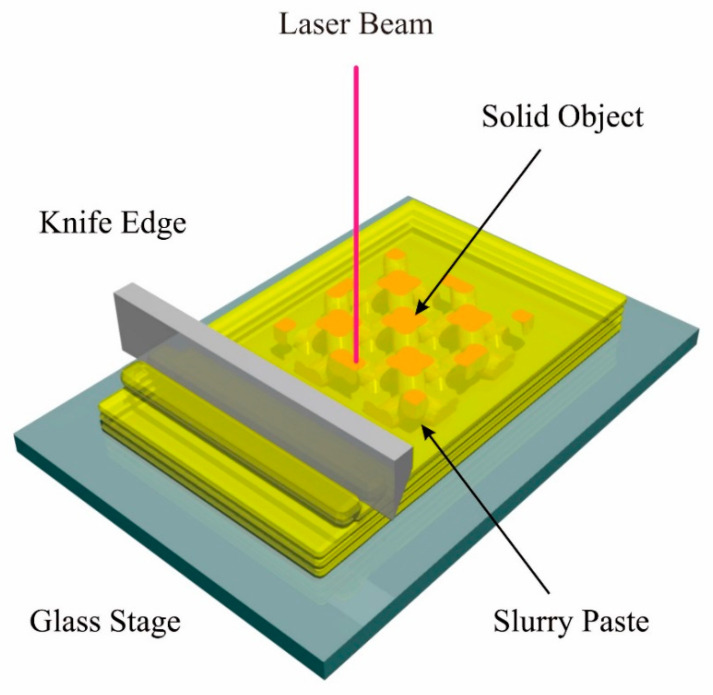
Schematic illustration of stereolithographic additive manufacturing (STL-AM). Cross-sectional layers are formed on paste materials by laser scanning. Solid objects are fabricated via layer laminating and interlayer bonding.

**Figure 2 materials-14-07090-f002:**
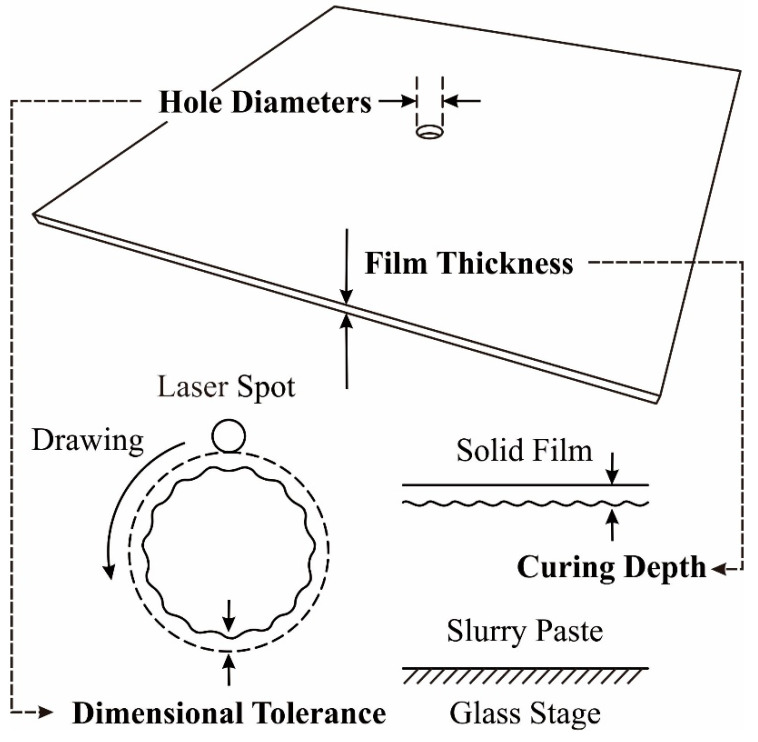
Drawing pattern of film specimens to optimize lithographic conditions. Hole diameters and layer thicknesses were measured by a digital optical microscope. Dimensional tolerances and curing depth were estimated.

**Figure 3 materials-14-07090-f003:**
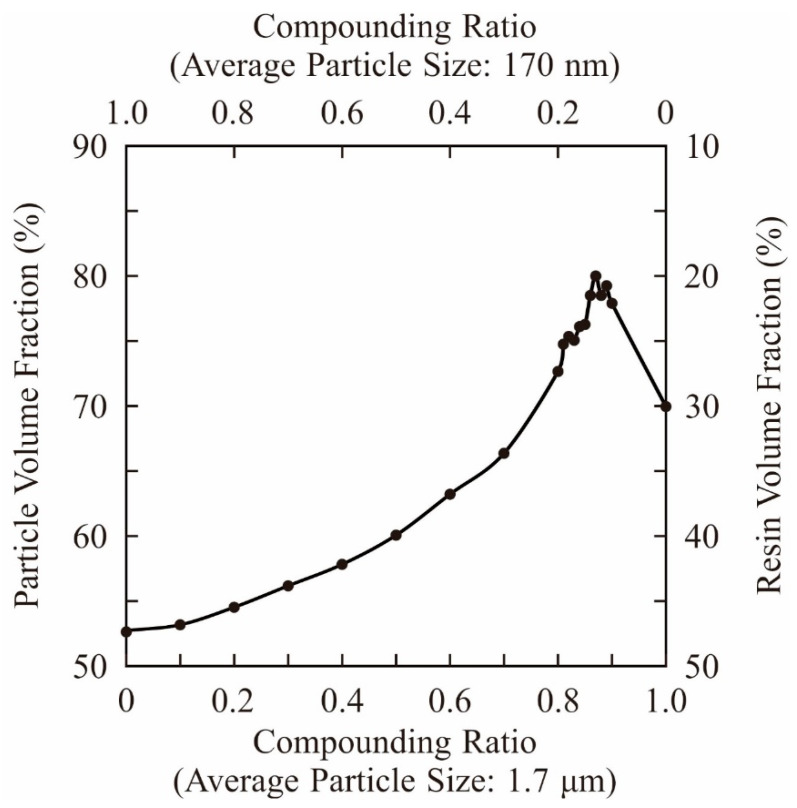
Volume fractions at compounding ratios. Binary particle sizes of 170 nm and 1.7 μm in diameters were adjusted. Closed packs of spherical fillers were simulated by a distinct element method (DEM).

**Figure 4 materials-14-07090-f004:**
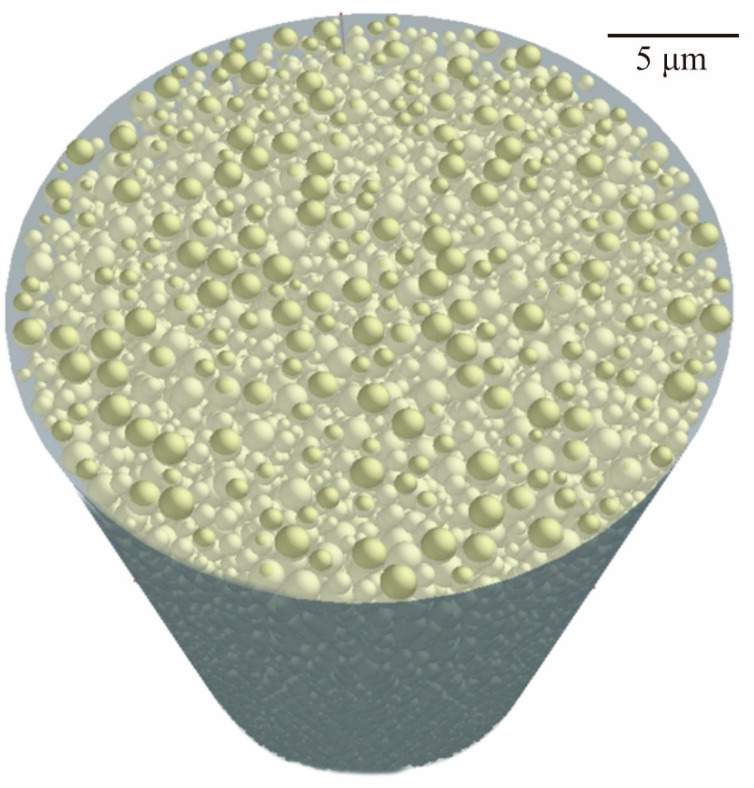
Dispersion profiles of bimodal powders in resin matrixes. Maximum contents were optimized at 80% of the total volume fraction. Computer graphics were plotted according to the DEM simulations.

**Figure 5 materials-14-07090-f005:**
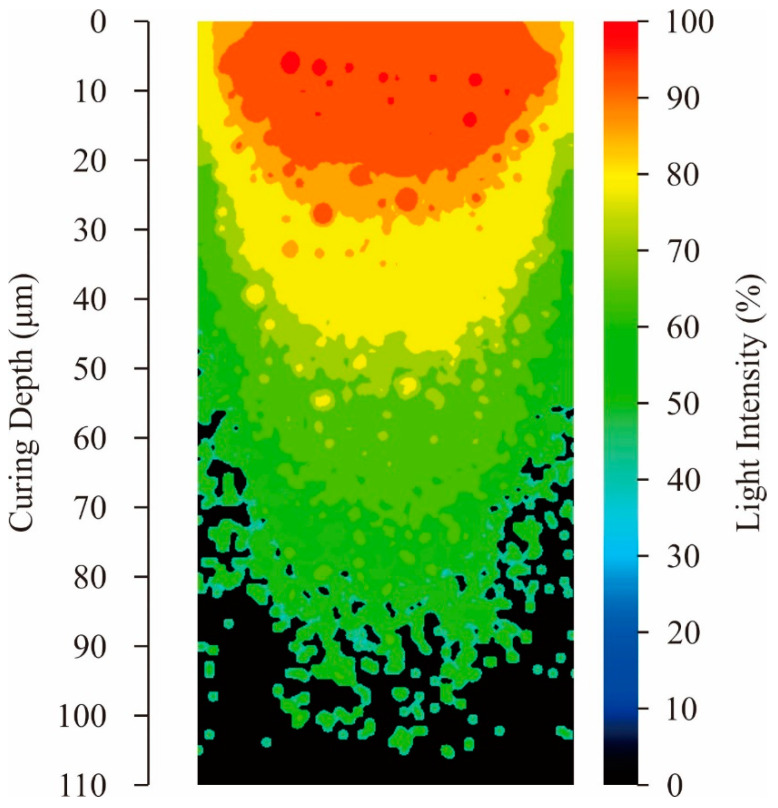
Intensity profile of optical irradiation at the laser spot. Vertical sections of polymerized regions were visualized. Ultraviolet (UV) light propagation in the particles gaps was simulated by the ray tracing (RT) method.

**Figure 6 materials-14-07090-f006:**
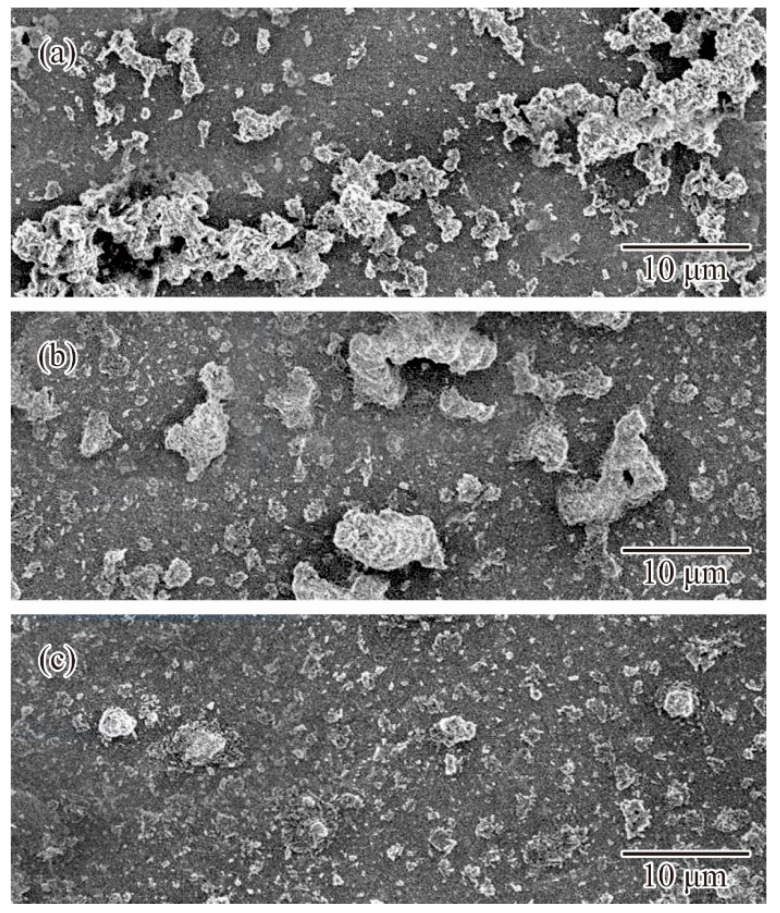
Alumina particles dispersed in acrylic resin. Paste containers were rotated and revolved for processing times of 300, 600, and 900 s. Cross sections of solid films are shown in (**a**–**c**).

**Figure 7 materials-14-07090-f007:**
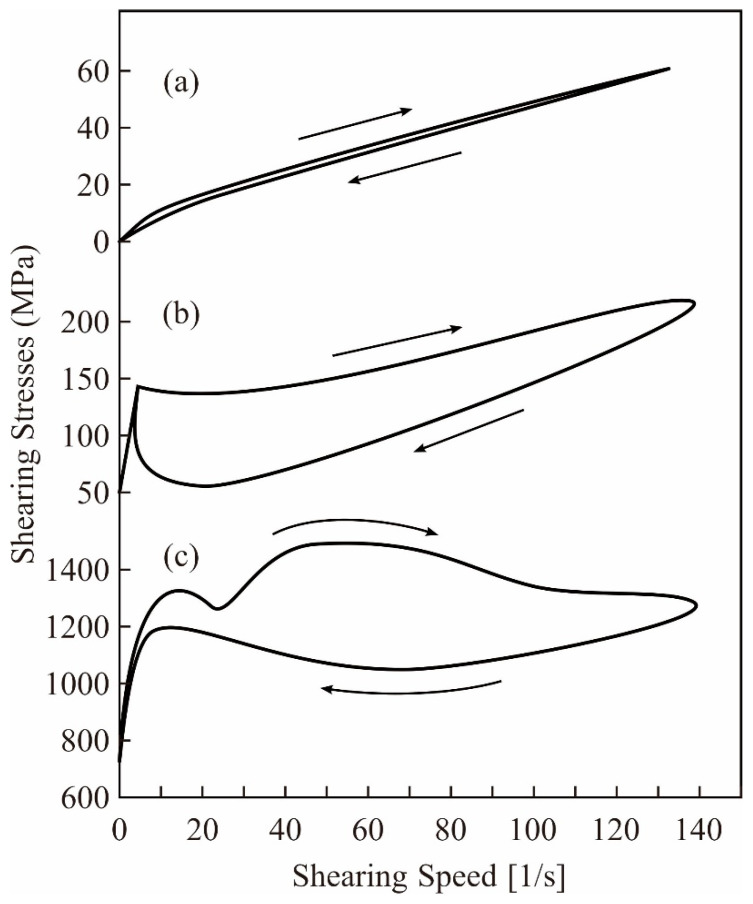
Dynamic profiles of shearing stresses and speeds. Rheological profiles were analyzed by a kinematic viscometer (KV). Viscose pastes processed for 300, 600, and 900 s are shown in (**a**–**c**), respectively.

**Figure 8 materials-14-07090-f008:**
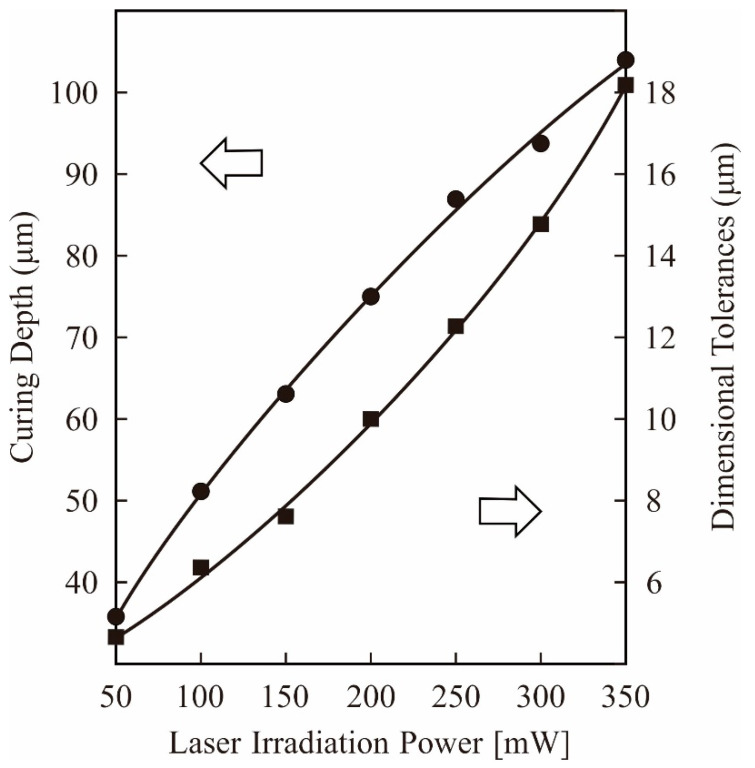
Corresponding values of the curing depth and dimensional tolerances for laser irradiation powers. Layer thickness and hole diameters were measured in the film specimens, as shown in Figure 2.

**Figure 9 materials-14-07090-f009:**
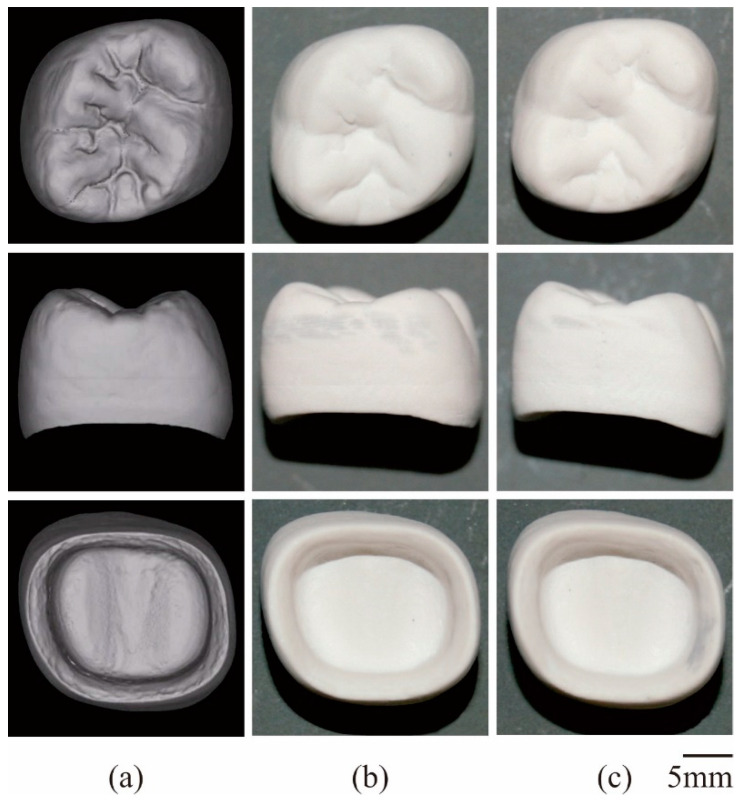
Dental crowns fabricated by STL-AM. Top and side views of the graphic model (**a**); composite precursor (**b**), and ceramic component (**c**) arranged in the same magnification.

**Figure 10 materials-14-07090-f010:**
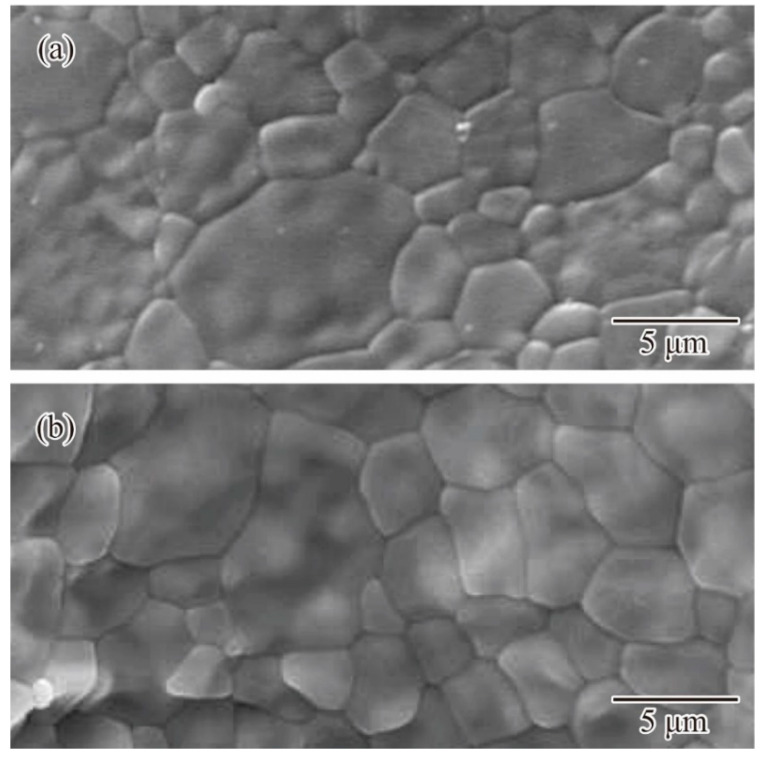
Ceramic microstructures in alumina crowns. Crystal grains observed by the scanning electron microscopy (SEM). Cross sections of the (**a**) horizontal and (**b**) vertical planes are defined as the parallel and vertical views of the laminated layers.

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
