# Peer review of "Systematic Compounding of Ceramic Pastes in Stereolithographic Additive Manufacturing"

_materials, 2021, doi:10.3390/ma14227090_

Round 1

Reviewer 1 Report

Dear author,

thanks for the interesting contribution on the highly relevant topic of ceramic AM for dental applications. For publishing this work in Materials the following revisions are suggested.

General remarks:

The state of the art of using AM of ceramics for dental applications should be briefly mentioned during the introduction.

The abstract says ‚coated as artificial enamelum‘, the conclusion ‘coated with artificial enamelum’.

Also, the process of applying the glass layers is not presented in the paper; please explain how this was conducted and which materials were used for this.

Concerning the characterization of the parts, please explain how many samples for used for the individual tests (hardness, strength, density).

Why was alumina used for printing crowns? In the field, mostly zirconia or glass ceramics are used for such applications.

You mention a strength of 800MPa; this would be an exceptionally high value for alumina. Can you please share the individual data points and corresponding Weibull statistics?

Specific remarks:

L56: what does ‘mechanically shaved teeth’ mean?

L104 & 106: is revolution the right English term here?

L108: maybe use ethanol instead of ethyl alcohol

L152: please explain what Is meant by polyhedral features

L153: please quantify the offset values

L157: This phrase is a bit redundant, laminated layers are already joined interlamellar.

L159: what is meant by composite precursors?

L162: What is the total duration of debinding and sintering? Please show the corresponding temperature protocol?

L233: The sentence starting with ‘In total…’ is not clear

L235: Please rephrase the sentence starting with ‘ Afterwards,…’ In the current form it sounds like this was a process step and not the result of a measurement. Also, please use ‘acrylic matrix’ instead of ‘acryl matrices’

L236: It is explicitly mentioned that no defects were found; but shouldn’t such defects not only occur during the dewaxing and sintering?

L247: does this mean that cracks or delaminations were found on other surfaces that were not smooth? Please clarify.

L270: This wording is ambiguous; does it mean that there were no defects found, or that these defects were not included in the shown images

Author Response

Dear Reviewer,

Thank you very much for your kind reviewing my review manuscript. I would like to reply for your interesting General Remarks and Specific Remarks as follows. I am looking forward to hearing your final decision.

Best regards,

Soshu Kirihara

Professor, JWRI, Osaka University, Japan

General Remarks

GR-1: The state of the art of using AM of ceramics for dental applications should be briefly mentioned during the introduction.

GR-2: The abstract says‚ “coated as artificial enamelum”, the conclusion “coated with artificial enamelum”.

GR-3: Also, the process of applying the glass layers is not presented in the paper; please explain how this was conducted and which materials were used for this.

GR04: Why was alumina used for printing crowns? In the field, mostly zirconia or glass ceramics are used for such applications.

Answer:  According to GR 1, 2, 3 and 4, the following sentences were added into “Introduction” from L-52. About artificial enamelum, related sentences were deleted from “Abstract” and “Condlusion”, because the surface treatment is not directly concerned with this review.

“Considering the rapid fabrication of biological implants, ceramic dental crowns can be successfully processed by using computer aided design and manufacturing (CAD/CAM) [10]. Calcinated zirconia bodies with disc shapes are automatically cut and shaped by top-down processing. Crown precursors can be sintered into fine ceramic components with mechanical properties that satisfy the practical strengths required by dental bridges. Recently, translucent zirconia with systematic yttria additions has been used to produce natural textures. In comparison, our research group has attempted to process alumina dental crowns by STL-AM using viscose resin pastes with dense par-ticle dispersions [11]. Functional features could be formed on inner crown surfaces via bottom-up processing to realize rigid fastening with artificial dental roots. The trans-lucent bodies could exhibit mechanical properties sufficient for single crown use.”

GR-5: Concerning the characterization of the parts, please explain how many samples for used for the individual tests (hardness, strength, density).

GR-6: You mention a strength of 800MPa; this would be an exceptionally high value for alumina. Can you please share the individual data points and corresponding Weibull statistics?

Answer:  At first, please accept our apology. The bending strength of 800 MPa was incorrect. The true value was revised into 480 MPa as shown in L-278. Moreover, the following sentences of measurement and estimate methods of mechanical properties were add at the end of “Procedures” from L-174.

“Plate specimens with dimensions of 20 mm × 5 mm × 1 mm were similarly fabricated. The mechanical strengths were measured by four-point bending tests, the micro-structural hardness was assessed by Vickers testing, and the relative densities were measured using the Archimedes method. The strength, hardness, and density were obtained from each specimen, which were arranged into groups of seven such that the highest and lowest measured values could be excluded and the average values calcu-lated from the remaining five data.”

Specific Remarks

L-67: What does “mechanically shaved teeth” mean?

Answer:  The sentence was changed as follows.

“On dental treatments, machined teeth should be covered with prosthetic appliances.”

L-114: Is revolution the right English term here?

Answer: The words of “rotation” and “revolution” were changed as “rotating” and “revolving”, respectively.

L-118: maybe use ethanol instead of ethyl alcohol

Answer: According you suggestion, “ethyl alcohol” was revised as “ethanol”.

L-160: Please explain what is meant by polyhedral features

Answer: The “polyhedral features” was changed into “polyhedrally approximated features”.

L-162: Please quantify the offset values

Answer: To clearly explain, the following sentence was revised.

“The polygonal outlines of the cross sections were slightly shifted inside to offset dimensional tolerances.”

L-166: This phrase is a bit redundant, laminated layers are already joined interlamellar.

Answer: I agree with our suggestion. The following sentence was deleted.

“Solid components were built by interlamellar joining of the laminated layers.”

L-169: what is meant by composite precursors?

Answer: “Composite precursors” was revised as “Obtained precursors”.

L172: What is the total duration of debinding and sintering? Please show the corresponding temperature protocol?

Answer: According to your suggestion, the following sentence was inserted.

“Total duration time of the heating, dewaxing, sintering, and cooling was 30 h.”

L-245: The sentence starting with ‘In total…’ is not clear

Answer: The first part was changed as follows.

“Solid layers of 50 in total were continuously laminated for 40 min in the process time.”

L-248: Please rephrase the sentence starting with ‘Afterwards,…’ In the current form it sounds like this was a process step and not the result of a measurement. Also, please use ‘acrylic matrix’ instead of ‘acryl matrices’

Answer: According to your suggestions, the sentence was revised as follows.

“Alumina particles were showing homogeneous dispersions in the acryl matrix.”

L-249: It is explicitly mentioned that no defects were found; but shouldn’t such defects not only occur during the dewaxing and sintering?

Answer: The sentence was revised as follows.

“Impressions of blowholes contaminations at paste spreading and delamination occurrences at layer bonding were not left in the microscopic field.”

L-255: Does this mean that cracks or delaminations were found on other surfaces that were not smooth? Please clarify.

Answer: The “smooth” was changed into “component”.

L-272: This wording is ambiguous; does it mean that there were no defects found, or that these defects were not included in the shown images

Answer: The “microscopic field” was changed into “shown images”.

Reviewer 2 Report

  • Can you explain what is the objective of your paper?
  • Can you provide a detailed explanation on the question on part measurement and accuracy evaluation?
  • The extensive use of self-references is clearly inadequate (19 out of 28 references were self-references).
  • Neither the objective nor the accuracy evaluation are clearly explained in the paper. Figures are poor and even contain groos errors (Figure 1).

    Consequently, my recommendation is that the paper should be rejected.

Author Response

Dear Reviser,

Thank you very much for your reviewing my manuscript. According to your suggestion, (1) my self-citation rate had been reduced at about 23%, and (2) the redundant caption was removed from Fig. 1. About object of this paper and part measurement and accuracy evaluation, (3) the third paragraph and the last three sentences were added to explain. If you have time for second round reviewing, could you please check the above-mentioned points of (1), (2) and (3)?

Best regards,

Soshu Kirihara (Osaka University, Japan)

Your Comments: Can you explain what is the objective of your paper? Can you provide a detailed explanation on the question on part measurement and accuracy evaluation? The extensive use of self-references is clearly inadequate (19 out of 28 references were self-references). Neither the objective nor the accuracy evaluation are clearly explained in the paper. Figures are poor and even contain groos errors (Figure 1). Consequently, my recommendation is that the paper should be rejected.

Reviewer 3 Report

This work reviews the stereolithographic additive manufacturing of ceramic pastes, addressing the design, manufacture and evaluation of dental crowns. It is an interesting work that highlights the potential of this technique for the manufacture of dental pieces, with results adequately presented and discussed. The work is suitable for publication in Materials, but the self-citations (18 of the 28 total citations) seem excessive. Although they validate the continuity and experience in this topic, please reconsider including only the most relevant and significant ones so that the percentage of self-citations does not exceed a healthy 15-20%. There are also a couple of details that could be corrected before publication: the legend that indicates the "knife edge" appears duplicated in figure 1 and there is a typographical error in the author's name in reference [23].

Author Response

Dear Reviewer,

Thank you very much for your kind reviewing my manuscript. I would like to report for you that I have revised the manuscript according to your comments and suggestions. I am looking forward to hearing from your suggestion on the second round.

Best regards,

Soshu Kirihara (Osaka University, Japan)

Comment 1/2: The work is suitable for publication in Materials, but the self-citations (18 of the 28 total citations) seem excessive. Although they validate the continuity and experience in this topic, please reconsider including only the most relevant and significant ones so that the percentage of self-citations does not exceed a healthy 15-20%.

Answer 1/2: According to your suggestion, I have reduced the number of self-citations at 3 in 13 references as total. The self-citation rate is about 23%. Could you please see the revised references list?

Comment 2/2: There are also a couple of details that could be corrected before publication: the legend that indicates the "knife edge" appears duplicated in figure 1 and there is a typographical error in the author's name in reference [23].

Answer 2/2: Thank you for your kind and prompt checking. I would like to say I am very sorry for my simple mistakes. Figure 1 was revised, and true author’s name was listed in reference [23]

Round 2

Reviewer 1 Report

Dear author,

thanks for the edits and the answers and clarifications to my previous comments. This revised version of the manuscript is considered appropriate for being published in Materials.

Reviewer 2 Report

OK. Thanks.